# Early Enteral Nutrition with High-Protein Whey Peptide Digestive Nutrients May Improve Prognosis in Subarachnoid Hemorrhage Patients

**DOI:** 10.3390/medicina58091264

**Published:** 2022-09-13

**Authors:** Kaima Suzuki, Hiroki Sato, Hiromi Mori, Ryosuke Matsumoto, Yoshihiro Arimoto, Hiroshi Sato, Tomoya Kamide, Toshiki Ikeda, Yuichiro Kikkawa, Hiroki Kurita

**Affiliations:** 1Department of Cerebrovascular Surgery, International Medical Center, Saitama Medical University, Hidaka, Saitama 350-0495, Japan; 2Department of Nutrition, International Medical Center, Saitama Medical University, Hidaka, Saitama 350-0495, Japan; 3Department of Medical Record Management, International Medical Center, Saitama Medical University, Hidaka, Saitama 350-0495, Japan; 4Department of Pharmacy, International Medical Center, Saitama Medical University, Hidaka, Saitama 350-0495, Japan; 5Division of Gastroenterological Surgery, International Medical Center, Saitama Medical University, Hidaka, Saitama 350-0495, Japan

**Keywords:** subarachnoid hemorrhage, enteral nutrition, oligomeric formula, outcome, modified rankin scale, diarrhea

## Abstract

*Background and Objectives*: Nutritional management in patients with subarachnoid hemorrhage (SAH) during the acute phase is important; however, there is no proper evidence or recommendations on the appropriate nutrients for early enteral nutrition. This study compared the influence the two different tube-feeding liquid diets for early enteral nutrition might have on the prognosis of patients with SAH. *Materials and Methods*: In a seven-year period, this single-center retrospective study included 245 patients with aneurysmal SAH who underwent craniotomy and aneurysm neck clipping and received enteral nutrition. The patients were divided into two groups according to the nutrient received: (1) high-protein whey peptide oligomeric formula diet (oligomeric group, 109 patients); and (2) high eicosapentaenoic acid-containing polymeric formula diet (polymeric group, 136 patients). The modified Rankin Scale (mRS) score at discharge was evaluated as the primary outcome. The presence or absence of diarrhea (watery stool and mushy stool) during the period from initiation of enteral nutrition to discharge from the stroke unit was also evaluated. *Results:* There were no significant differences in patient characteristics between groups. The time until initiation of enteral feeding in the oligomeric and polymeric groups was 2.8 ± 2.3 and 2.9 ± 2.2 days, respectively. The proportion of patients with mRS scores of 0–1 was significantly higher in the oligomeric group (25.7%) than in the polymeric group (14.7%) (*p* = 0.036), while the incidence of watery stool was significantly lower in the oligomeric group (15.8% to 34.3% in the polymeric group) (*p* = 0.003). Multivariate analyses confirmed that the oligomeric diet and the presence or absence of diarrhea significantly affected the mRS scores. *Conclusions**:* The adoption of early enteral nutrition with high-protein whey peptide digestive nutrients might be associated with superior mRS scores at discharge and decreased diarrhea in patients with SA, indicating that the choice of nutrients might affect the outcome and prognosis.

## 1. Introduction

Stroke is the second-leading cause of death, and the third-leading cause of death and disability combined in 2019 globally. In 2019, a total of 6.55 million individuals died of strokes worldwide [1]. Among patients with strokes, the prognosis of patients with subarachnoid hemorrhage (SAH) due to ruptured cerebral aneurysms is particularly poor, and approximately 20% patients with SAH need help to perform the activities of daily living [2].

A craniotomy for the aneurysm neck clipping is an advanced, and sometimes high-risk procedure [3]; and nutritional requirements are higher than that in normal circumstances or after coiling [4]. In SAH patients, the daily demand for proteins increases significantly in the acute phase due to increased anabolism and catabolism, and stress coefficients for the necessary calories’ calculation are anticipated to be high [5,6]. Malnutrition during the post-stroke acute phase is an independent factor for poor outcome [7], and a multicenter study found that SAH patients received under 60% of their prescribed caloric and protein intake [8]. According to a report in Japan, 44% patients with cerebral infarction and 100% patients with SAH who were transferred from acute-phase hospitals to recovery-phase hospitals reportedly had a nutritional disorder, and even a specific score was recently introduced to assess the correlation between nutrition and outcomes [9,10].

During the acute phase in patients with stroke, including those with SAH, early enteral nutrition is recommended [11,12,13], especially for the poor-grade patients [14]. In clinical practice, various types of liquid diets are used for enteral nutrition and are generally classified into a polymeric formula or an oligomeric formula based on the nitrogen source. For the oligomeric formula, the source of nitrogen is amino acids or peptides; hence, they are considered to be absorbed better than the polymeric formula, in which the source of nitrogen is protein [15,16].

Although the importance of nutritional management for patients with stroke, including those with SAH, during the acute phase is evident, as described above, there is no evidence for the selection of appropriate nutrients for early enteral nutrition. Hence, this study aimed to investigate the influence of different tube-feeding liquid diets on the outcome of patients with SAH.

## 2. Materials and Methods

### 2.1. Study Design

This was a retrospective cohort study of patients with SAH, due to intracranial aneurysm rupture, who underwent craniotomy and neck clipping between 1 April 2012 and 31 March 2019 at the Department of Cerebrovascular Surgery of the International Medical Center, Saitama Medical University.

For the patient characteristics, age on admission, sex, body weight, presence or absence of comorbidities (atrial fibrillation, hypertension, diabetes mellitus, hyperlipidemia, and bleeding complication), data on laboratory tests (serum albumin, Na, K, blood glucose, and lymphocyte count), Hunt and Kosnik grading (HK), and the Fisher classification were collected from the medical records.

A polymeric formula diet was used for patients before 2016, while an oligomeric formula diet was used for patients from 2016 onwards. Data on the types of enteral nutrition products, the modified Rankin Scale (mRS) scores at discharge, and the presence or absence of diarrhea (watery stool and mushy stool) during the period from the start of enteral nutrition to discharge from the stroke care unit, as well as the duration of hospital stay and outcome indices were collected.

### 2.2. Patients

Of the 407 patients who underwent craniotomy and surgical aneurysm occlusion within 24 h after the onset of SAH due to ruptured aneurysm, 291 patients received enteral tube feeding according to the early nutritional management protocol (Figure 1).

Patients who did not receive any of the two nutrients under evaluation (discussed below) (42 patients) and patients who received both nutrients (four patients) were excluded. The data of 245 patients were analyzed, and they were compared by classifying them into two groups based on the nutrients administered: 109 patients who received a high-protein whey peptide-containing oligomeric formula (Peptamen AF^®^; Nestlé Japan Ltd., Nestlé Health Science, Tokyo, Japan) were classified into the oligomeric group; and 136 patients who received a high eicosapentaenoic acid-containing polymeric formula (Oxipa^®^; ABBOT Japan Co., Ltd., Tokyo, Japan) were placed in the polymeric group (Figure 2).

### 2.3. Statistical Analyses

Patients’ background characteristics and outcomes between the groups were analysed using the Mann–Whitney U test for continuous variables and the Fisher’s exact probability test for categorical variables. For mRS scores at discharge, the distributions of grades 0–6 were compared between the groups. In addition, categories of mRS scores 0–1 and 2–6, and categories of mRS scores 0–2 and 3–6, were compared between the groups. The incidence of diarrhea (including both watery and mushy stool, and only watery stool) was compared between the groups using an analysis set defined as patients whose data could be retrospectively collected. When there was a difference in the categorized mRS scores between the groups, univariate logistic regression analyses were performed for the mRS categories, and differences showing patients background characteristics, as well as for the types of nutrients, to examine the factors independently affecting the mRS scores.

Furthermore, to evaluate the influence of different nutrients adjusted for the effect of background characteristics on mRS scores, multivariate logistic regression analyses were performed with mRS scores (category with a difference) as the objective variable, and nutrients, sex, age, HK, and preoperative body weight as explanatory variables. Diarrhea until discharge from the stroke care unit, which could theoretically be an intermediate variable of nutritional intervention for mRS scores, was excluded from the explanatory variables. Logistic regression analyses were separately performed in the presence or absence of diarrhea, sex, age, HK, and preoperative body weight as explanatory variables to assess the effect of the presence or absence of diarrhea on mRS scores. A *p*-value of <0.05 was considered statistically significant. SAS 9.4 (SAS Institute, Inc., Cary, NC, USA) was used for statistical analyses.

## 3. Results

There were no significant differences in the patients’ background characteristics between the groups (Table 1).

There was no difference in the distributions of mRS scores (*p* = 0.826); however, the outcomes of good scores (mRS score 0–1) were different between the groups (Table 2). The proportion of patients with mRS scores 0–1 was significantly higher in the oligomeric group (25.7%) than in the polymeric group (14.7%) (*p* = 0.036).

There was no difference in the duration of hospital stay between the two groups.

Due to the difference in the proportion of patients with mRS scores 0–1 between the oligomeric and polymeric groups, factors contributing to the outcome of mRS scores 0–1 were analyzed (Table 3). Univariate regression analyses revealed that age, preoperative body weight, and HK significantly affected mRS scores 0–1 in addition to the oligomeric diet.

The results of multivariate logistic regression analyses showed a significant positive influence of the administered enteral nutrition diet on the mRS scores 0–1 in the oligomeric group (*p* = 0.023; OR: 2.447; 95% CI: 1.134–5.284) (Table 4).

In addition, the presence or absence of diarrhea significantly affected the mRS scores 0–1 (*p* = 0.047; OR: 0.41; 95% CI: 0.167–0.988) in a multivariate analysis (Table 5).

## 4. Discussion

This study investigated the influence of different tube-feeding liquid diets used for early enteral nutrition on the prognosis of patients with SAH.

The results demonstrated that the percentage of patients with mRS scores 0–1 at discharge was significantly higher in the oligomeric group than in the polymeric group. Furthermore, the results of regression analyses suggested that the use of the oligomeric formula diet had a positive effect on the prognosis. In addition, the incidence of diarrhea was lower in the oligomeric group than in the polymeric group.

A comparison of the compositions of liquid diets used in accordance with the protocols in this study revealed that the amount of protein per day was 10.2 g higher in the liquid oligomeric formula diet (Figure 2 and Table 1). The source of nitrogen in the liquid oligomeric formula diet is peptides, and this diet is considered to be absorbed better than the liquid polymetric formula diet, in which the nitrogen source is proteins [15,16]. Moreover, in this study, a whey peptide was used as the source of nitrogen in the liquid oligomeric formula diet, which has been reported to be particularly well absorbed [17,18,19]. In contrast, casein was the primary protein source in the liquid polymeric formula diet, and its absorption is considered poor [20].

In this study, the proportion of patients with mRS scores 0–1 was higher in the oligomeric group than in the polymeric group, and the proportion of patients with mRS scores 0–2 was also higher in the oligomeric group. In addition, the regression analyses suggested that the use of oligomeric formula had positively affected the achievement of mRS scores 0–1. Based on the previously reported post-stroke malnutrition prognosis affection, the absorption of an adequate amount of proteins has been proposed to maintain the nutritional status after craniotomy for aneurysm occlusion in patients with SAH during the phase of increased anabolism and catabolism [4], leading to a significantly higher proportion of patients with mRS scores of 0–1 [7].

In this study, the incidence of diarrhea was lower in the oligomeric group. Diarrhea can occur for many reasons in patients with SAH, and it may be related to the tube-feeding nutrition as well [21]. The frequency of tube-feeding–induced diarrhea is influenced by dosage, dosing rate, osmotic pressure of the liquid diet, fat quantity, and absorption [22].

In this study, the osmotic pressure of the oligomeric formula diet and polymeric formula diet was 440 and 384 mOsm/L, respectively, while the fat mass was 39.6 and 70.2 g/day respectively. The cause for the difference in the incidence of diarrhea in the present study is likely due to the absorbability of the nitrogen source (whey peptide) and the difference in fat mass. In addition, the presence or absence of diarrhea affected the mRS scores of 0–1 in this study. Diarrhea may cause exhaustion, dehydration, and malnutrition, which may all affect the overall mRS score [21].

There was a significant difference in the proportion of patients with mRS scores 0–1 due to the nutrient differences, but there was no difference in the distribution of mRS scores (Table 2). Evaluation of the distribution by the grade of mRS revealed that the proportions of patients with mRS scores 0 and mRS scores ≥ 4 were identical between the two groups, whereas the proportion of patients with initial mRS scores of 1–3 decreased in the oligomeric group but increased in the polymeric group. Based on these results, it can be inferred that, although our sample size was small, the administration of a high-protein whey peptide diet might improve the outcome of patients with a moderate prognosis, such as those with mRS scores 2 or 3 (i.e., from mRS 3 to 2 and mRS 2 to 1). However, the difference in nutritional intervention might have a lesser impact on patients with unfavorable mRS scores (i.e., mRS 4–6), and on patients with an mRS score of 0 because their prognosis might have been good regardless of the nutritional intervention. The proportion of patients with mRS scores 0–1 by the severity (HK) of SAH is shown in Table 2. The proportion of those with mRS scores 0–1 was significantly higher among the oligomeric group patients with moderate severity scores (HK of 2–4), thus confirming that the positive effects of different nutrients were more likely to affect patients with moderately severe HK.

The study demonstrated a statistically significant correlation between the enteral nutrition given and the overall outcome mRS score and the occurrence of diarrhea. Although the study period was relatively long, there were no significant treatment changes other than the difference in nutrition formula during the study period. These results encourage the use of a high-protein whey peptide-containing oligomeric formula in patients with SAH due to the potentially beneficial effects.

This study has several limitations. The retrospective observational study design with the limited amount of usable data might not have included potentially significant characteristics, and changes in nutritional status, such as body weight and other parameters. Additional studies are necessary to understand the influence of protein absorption and the inhibition of diarrhea on mRS scores based on the administration of different nutrients. This study compared only two nutrient products, and further studies on the influence of different nutritional compositions on the outcomes are necessary. Apart from the differences in the nitrogen source and amount of protein, the number of calories might also vary between the diets, and may be significant.

## 5. Conclusions

The use of early enteral nutrition with high-protein whey peptide digestive nutrients in patients with SAH decreases the occurrence of diarrhea, and might be associated with superior mRS scores at discharge. Although further prospective research is needed to deliver clear recommendations and guidelines for the nutritional support of these complex patients, knowledge on the selection of appropriate nutrients suitable for the patients’ conditions is definitely an important element of stroke care, and stroke specialists should be aware of it.

## Figures and Tables

**Figure 1 medicina-58-01264-f001:**
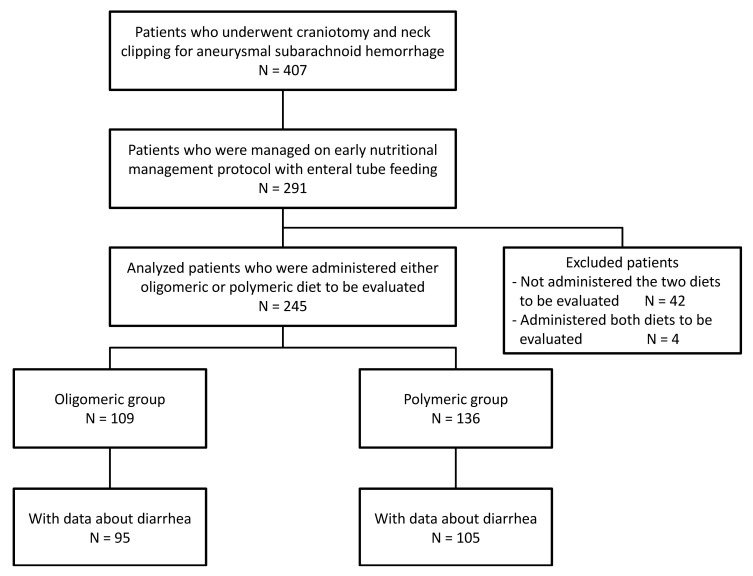
Disposition of patients.

**Figure 2 medicina-58-01264-f002:**
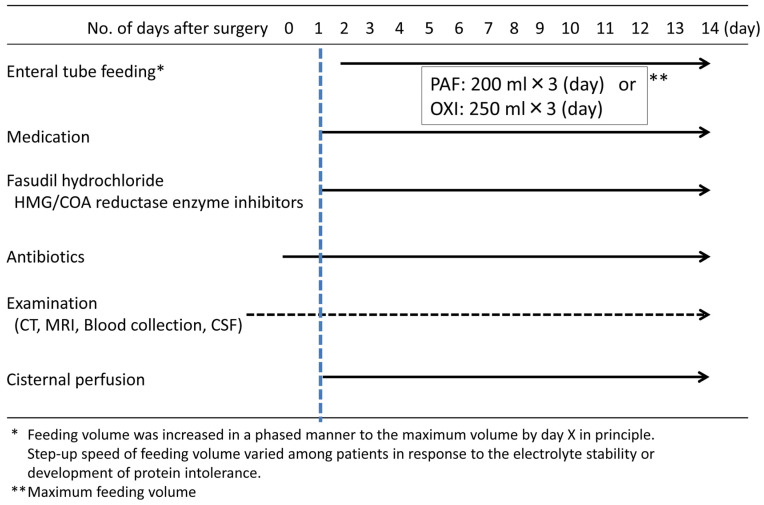
Treatment protocol for subarachnoid hemorrhage.

**Table 1 medicina-58-01264-t001:** Patients’ background characteristics.

	Oligomeric Group	Polymeric Group	*p*-Value
	*n* = 109	*n* = 136	
Female, *n* (%)	66 (42.6)	89 (57.4)	0.505
Age, (years) mean ± SD	65.2 ± 13.8	63.1 ± 12.6	0.209
Body weight, (kg) mean ± SD	59.3 ± 13.2	55.9 ± 10.4	0.088
Comorbidities, *n* (%)			
Atrial fibrillation	1 (0.9)	1 (0.7)	1
Hypertension	32 (29.4)	27 (19.9)	0.099
Diabetes mellitus	7 (6.4)	3 (2.2)	0.114
Hyperlipidemia	7 (6.4)	2 (1.5)	0.082
Bleeding complication	8 (7.3)	13 (9.6)	0.648
HK grade, *n* (%)			
Grade 1	14 (12.8)	6 (4.4)	0.933
Grade 2	24 (22.0)	42 (30.9)	
Grade 3	22 (20.2)	24 (17.6)	
Grade 4	25 (22.9)	47 (34.6)	
Grade 5	24 (22.0)	17 (12.5)	
Fisher classification, *n* (%)			
I	0 (0.0)	2 (100)	0.536
II	11 (64.7)	6 (35.3)	
III	73 (43.5)	95 (56.5)	
IV	25 (43.9)	32 (56.1)	
Albumin, (g/dl) mean ± SD	4.0 ± 0.6	4.0 ± 0.7	0.638
Na, (mEq/L) mean ± SD	139.8 ± 3.7	139.4 ± 3.1	0.42
K, (mEq/L) mean ± SD	3.6 ± 0.5	3.5 ± 0.4	0.069
Blood glucose, (mg/dl) mean ± SD	169.6 ± 53.5	173.1 ± 49.5	0.606
Lymphocyte count, mean ± SD	2318.3 ± 1820.8	1923.2 ± 1550.2	0.072
Time to enteral nutrition (day), mean ± SD	2.8 ± 2.3	2.9 ± 2.2	0.783
Oligomeric group, Peptamen AF group; Polymeric group, Oxipa group; HK grade, Hunt and Kosnik grading; Na, sodium; K, potassium; SD, standard deviation.			

**Table 2 medicina-58-01264-t002:** Nutrients, mRS, and presence or absence of diarrhea.

		Oligomeric Group	Polymeric Group	*p*-Value
		*n* = 109	*n* = 136	
mRS at discharge, *n* (%)	**0**	7 (6.4)	9 (6.6)	0.826
	1	21 (19.3)	11 (8.1)	
	2	12 (11.0)	22 (16.2)	
	3	9 (8.3)	23 (16.9)	
	4	24 (22.0)	30 (22.1)	
	5	33 (30.3)	39 (28.7)	
	6	3 (2.8)	2 (1.5)	
mRS 0–1, *n* (%)		28 (25.7)	20 (14.7)	0.036
Diarrhea, *n* (%)		43 (45.3)	70 (66.7)	0.003
Watery stool, *n* (%)		15 (15.8)	36 (34.3)	0.003
Duration of hospital stay, (day), mean ± SD		54.8 ± 28.6	50.9 ± 22.6	0.237
mRS, modified Rankin Scale; Oligomeric group, Peptamen AF group; Polymeric group, Oxipa group; SD, standard deviation.				

The incidence of diarrhea was significantly lower in the oligomeric group (45.3%) than in the polymeric group (66.7%) (*p* = 0.003). The incidence of watery stool was also significantly lower in the oligomeric group (15.8%) than in the polymeric group (34.3%) (*p* = 0.003).

**Table 3 medicina-58-01264-t003:** Effect of enteral nutrition on prognosis (univariate analysis).

Evaluated Factor	mRS 0–1	mRS ≥ 2	Odds Ratio and 95% CI			
	*n* = 48	*n* = 197	Odds Ratio	Lower	Upper	*p*-Value
Enteral nutrition, *n* (%)						
Oxipa	20 (14.7)	116 (85.3)				0.033
Peptamen	28 (25.7)	81 (74.3)	2.005	1.057	3.803	
Albumin (g/dl)	42 (19.5)	173 (80.5)	1.42	0.7	2.884	0.331
Sex, *n* (%)						
Male	18 (20)	72 (80)				0.902
Female	30 (19.4)	125 (80.6)	0.96	0.5	1.843	
HK, *n* (%)						
Grade < III	29 (33.7)	57 (66.3)				<0.0001
Grade ≥ III	19 (11.9)	140 (88.1)	0.267	0.139	0.514	
Age category (years), *n* (%)						
<65	32 (29.1)	78 (70.9)				0.001
65 £	16 (11.9)	119 (88.1)	0.328	0.169	0.637	
Age (cont.)	48 (19.6)	197 (80.4)	0.955	0.932	0.979	0.0002
Body weight (cont.)	43 (19.6)	176 (80.4)	1.038	1.008	1.068	0.012

**Table 4 medicina-58-01264-t004:** Effect of enteral nutrition on prognosis (multivariate analysis).

Evaluated Factor	mRS 0–1	mRS ≥ 2		Odds Ratio 95% CI		
	*n* = 43	*n* = 176	Odds Ratio	Lower	Upper	*p*-Value
Enteral nutrition, *n* (%)						
Oxipa	20 (15.2)	112 (84.8)				0.023
Peptamen	23 (26.4)	64 (73.6)	2.447	1.134	5.284	
Sex, *n* (%)						
Male	16 (20.3)	63 (79.7)				0.095
Female	27 (19.3)	113 (80.7)	2.175	0.873	5.417	
HK, *n* (%)						
Grade < III	27 (36.0)	48 (64.0)				<0.0001
Grade ≥ III	16 (11.1)	128 (88.9)	0.146	0.066	0.325	
Age category (years), *n* (%)						
<65	28 (28.3)	71 (71.7)				0.008
65 £	15 (12.5)	105 (87.5)	0.327	0.143	0.749	
Body weight (cont.)	43 (19.6)	176 (80.4)	1.047	1.008	1.089	0.022

**Table 5 medicina-58-01264-t005:** Effect of diarrhea on prognosis (multivariate analysis).

	mRS 0–1	mRS ≥ 2	Odds Ratio and 95% CI			
	*n* = 31	*n* = 146	Odds Ratio	Lower	Upper	*p*-Value
Diarrhea, *n* (%)						
No	17 (26.2)	48 (73.8)				0.047
Yes	14 (12.5)	98 (87.5)	0.406	0.167	0.988	
Sex, *n* (%)						
Male	11 (18.3)	49 (81.7)				0.163
Female	20 (17.1)	97 (82.9)	2.139	0.734	6.231	
HK, *n* (%)						
Grade < III	19 (33.9)	37 (66.1)				0.0001
Grade ≥ III	12 (9.9)	109 (90.1)	0.132	0.05	0.347	
Age category (years), *n* (%)						
<65	19 (25.7)	55 (74.3)				0.051
65 £	12 (11.7)	91 (88.3)	0.392	0.153	1.004	
Body weight (cont.)	31 (17.5)	146 (82.5)	1.067	1.017	1.119	0.008

## Data Availability

All data generated or analyzed during this study are included in this article. Further enquiries can be directed to the corresponding author.

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
