# Peer review of "Early Enteral Nutrition with High-Protein Whey Peptide Digestive Nutrients May Improve Prognosis in Subarachnoid Hemorrhage Patients"

_medicina, 2022, doi:10.3390/medicina58091264_

Round 1
Reviewer 1 Report
This study investigated the influence of different tube-feeding liquid diets used for 152 early enteral nutrition on the prognosis of patients with SAH. The results demonstrated that the percentage of patients with mRS scores 0–1 at discharge was significantly higher in the oligomeric group than in the polymeric group. Furthermore, the results of regression analyses suggested that the use of the oligomeric formula diet had a positive effect on the prognosis.
As a hemorrhagic stroke patients are experiencing turbulent and intensive period after emergency surgery, it is important to perform all the necessary measures to ensure the best possible outcome. One of those measures is nutrition, which is often neglected. This manuscript presents important findings regarding the nutrition of these patients and guides the approach so that the outcome could be better in the future.
I suggest the authors do:
- rewrite the parts of abstract, especially to better define neck clipping (21) as aneurysm neck clipping or open surgery aneurysm treatment, etc.
- remove the April and March from abstract, and to rewrite the study period as: in a seven-year period.
- Better define the sentences in 47-50
Author Response
Dear Reviewer 1:
We thank for the valuable comments. Please find our responses below.
This study investigated the influence of different tube-feeding liquid diets used for 152 early enteral nutrition on the prognosis of patients with SAH. The results demonstrated that the percentage of patients with mRS scores 0–1 at discharge was significantly higher in the oligomeric group than in the polymeric group. Furthermore, the results of regression analyses suggested that the use of the oligomeric formula diet had a positive effect on the prognosis.
As a hemorrhagic stroke patients are experiencing turbulent and intensive period after emergency surgery, it is important to perform all the necessary measures to ensure the best possible outcome. One of those measures is nutrition, which is often neglected. This manuscript presents important findings regarding the nutrition of these patients and guides the approach so that the outcome could be better in the future.
I suggest the authors do:
- rewrite the parts of abstract, especially to better define neck clipping (21) as aneurysm neck clipping or open surgery aneurysm treatment, etc.
We rewritten the relevant parts of the Abstract.
- remove the April and March from abstract, and to rewrite the study period as: in a seven-year period.
We revised, and as you proposed, left the statement: "in a seven-year period."
- Better define the sentences in 47-50
The sentences on the Craniotomy and daily protein demand are clarified.
Reviewer 2 Report
I have reviewed with interest the manuscript entitled: "Early enteral nutrition with high-protein whey peptide digestive nutrients may improve prognosis in subarachnoid hemorrhage patients". This manuscript is very interesting since it deals with nutritional aspects in patients with SAH.
Some minor points:
1. I suggest the authors expand the discussion a little more
2. Check the grammar and syntax again.
Author Response
Dear Reviewer 2:
Thank you for your comments. We have revised the manuscript accordingly.
I have reviewed with interest the manuscript entitled: "Early enteral nutrition with high-protein whey peptide digestive nutrients may improve prognosis in subarachnoid hemorrhage patients". This manuscript is very interesting since it deals with nutritional aspects in patients with SAH.
Some minor points:
1. I suggest the authors expand the discussion a little more
The discussion is extended in general, with clarified statements, and regarding the implications of the study results.
2. Check the grammar and syntax again.
The manuscript was revised for English language related grammar and syntax.